# Association of *TNF-α* (-308G/A) Gene Polymorphism with Circulating TNF-α Levels and Excessive Daytime Sleepiness in Adults with Coronary Artery Disease and Concomitant Obstructive Sleep Apnea

**DOI:** 10.3390/jcm10153413

**Published:** 2021-07-31

**Authors:** Afrouz Behboudi, Tilia Thelander, Duygu Yazici, Yeliz Celik, Tülay Yucel-Lindberg, Erik Thunström, Yüksel Peker

**Affiliations:** 1Division of Biomedicine, School of Heath Sciences, University of Skövde, SE 54128 Skövde, Sweden; tilia.thelander@gmail.com; 2Koc University Research Center for Translational Medicine (KUTTAM), Koc University Hospital, TR 34010 Istanbul, Turkey; dyazici17@ku.edu.tr (D.Y.); ycelik19@ku.edu.tr (Y.C.); yuksel.peker@lungall.gu.se (Y.P.); 3Department of Dental Medicine, Karolinska Institute, SE 17177 Stockholm, Sweden; tulay.lindberg@ki.se; 4Department of Molecular and Clinical Medicine, Sahlgrenska Academy, University of Gothenburg, SE 40530 Gothenburg, Sweden; erik.thunstrom@vgregion.se; 5Department of Clinical Sciences, Respiratory Medicine and Allergology, Faculty of Medicine, Lund University, SE 22185 Lund, Sweden; 6Division of Pulmonary, Allergy, and Critical Care Medicine, University of Pittsburgh School of Medicine, Pittsburgh, PA 15213, USA

**Keywords:** coronary artery disease, obstructive sleep apnea, tumor necrosis factor

## Abstract

Obstructive sleep apnea (OSA) is common in patients with coronary artery disease (CAD), in which inflammatory activity has a crucial role. The manifestation of OSA varies significantly between individuals in clinical cohorts; not all adults with OSA demonstrate the same set of symptoms; i.e., excessive daytime sleepiness (EDS) and/or increased levels of inflammatory biomarkers. The further exploration of the molecular basis of these differences is therefore essential for a better understanding of the OSA phenotypes in cardiac patients. In this current secondary analysis of the Randomized Intervention with Continuous Positive Airway Pressure in CAD and OSA (RICCADSA) trial (Trial Registry: ClinicalTrials.gov; No: NCT 00519597), we aimed to address the association of tumor necrosis factor alpha (*TNF-α*)-308G/A gene polymorphism with circulating TNF-α levels and EDS among 326 participants. CAD patients with OSA (apnea–hypopnea-index (AHI) ≥ 15 events/h; *n* = 256) were categorized as having EDS (*n* = 100) or no-EDS (*n* = 156) based on the Epworth Sleepiness Scale score with a cut-off of 10. CAD patients with no-OSA (AHI < 5 events/h; *n* = 70) were included as a control group. The results demonstrated no significant differences regarding the distribution of the *TNF-α* alleles and genotypes between CAD patients with vs. without OSA. In a multivariate analysis, the oxygen desaturation index and TNF-α genotypes from GG to GA and GA to AA as well as the *TNF-α-308A* allele carriage were significantly associated with the circulating TNF-α levels. Moreover, the *TNF-α-308A* allele was associated with a decreased risk for EDS (odds ratio 0.64, 95% confidence interval 0.41–0.99; *p* = 0.043) independent of age, sex, obesity, OSA severity and the circulating TNF-α levels. We conclude that the *TNF-α-308A* allele appears to modulate circulatory TNF-α levels and mitigate EDS in adults with CAD and concomitant OSA.

## 1. Introduction

Coronary artery disease (CAD) is the most common type of heart disease and is characterized by a poor prognosis and a high risk of morbidity and mortality [1]. The traditionally recognized risk factors for CAD are age, male sex, unhealthy lifestyle, hypertension, diabetes and hyperlipidemia. There are also data suggesting that the interaction between genetic and environmental factors plays an important role in the development of CAD [2,3].

Obstructive sleep apnea (OSA) is a common disorder characterized by repeated upper-airway collapse during sleep, resulting in intermittent hypoxia, fragmented sleep, fluctuations in blood pressure and increased sympathetic nervous system activity [4]. The prevalence of OSA among CAD patients is very high (50% as compared to 10–20% in the general adult population) [5], and OSA patients have an increased risk for the development of CAD compared to individuals without OSA [6].

Increased inflammatory activity plays a crucial role in the development of atherosclerotic plaques and CAD [7]. Moreover, circulating levels of inflammatory markers predict future cardiovascular events, both in the general population [8] and in adults with a known cardiovascular disease [9]. OSA has also been associated with increased inflammatory activity, probably due to intermittent hypoxia and oxidative stress resulting in increased concentrations of free radicals [10]. Elevated levels of high-sensitivity C-reactive protein (hs-CRP), interleukin (IL)-6 and tumor necrosis factor (TNF)-α have been found in patients with OSA [11]. In longitudinal observational studies, the treatment of OSA with continuous positive airway pressure (CPAP) has been shown to normalize the levels of circulating inflammatory markers, supporting the link between systemic inflammation and OSA [12]. However, a growing number of studies suggest that inflammation can be a predisposing factor for sleep-related breathing disorders [13,14,15] rather than only the consequence of these conditions.

TNF-α is a pro-inflammatory cytokine that plays an important role in the immune system and the development of autoimmune and infectious diseases as well as atherosclerosis and CAD [16]. TNF-α is also involved in sleep regulation [17]. The existing literature has suggested elevated levels of circulating TNF-α in patients with OSA [18]. There is also increasing evidence for the involvement of genetic and environmental factors in the development of OSA [19].

The manifestation of OSA varies significantly between individuals in clinical cohorts, and not all OSA patients demonstrate the same set of symptoms; i.e., excessive daytime sleepiness (EDS) and/or increased levels of inflammatory biomarkers. The further exploration of the molecular basis of these differences is therefore essential for a better understanding of the characteristics, prognosis and cardiovascular outcomes in CAD patients with OSA.

Several potential genetic risk factors for OSA have been studied using SNP (single nucleotide polymorphism) analysis, among which TNF-α has received special attention [17,20]. There is an SNP (Rs1800629) in the promoter region of the *TNF-α* (position-308G/A), where allele A at this position (*TNF-α*-308A) is suggested to be associated with the higher occurrence of OSA [21] as well as with the severity of this disorder [18,22,23,24,25]. An association between *TNF-α*-308A allele and obesity is also reported [17,21], whereas there is no significant association of TNF-α-308G/A polymorphism with CAD according to a recent meta-analysis [26].

We have previously conducted studies in the Randomized Intervention with CPAP in Coronary Artery Disease and Sleep Apnea (RICCADSA) trial with the primary goal of addressing the impact of CPAP on cardiovascular outcomes in revascularized CAD patients with OSA [27,28]. In the current work, we hypothesize that the *TNF-α*-308G/A gene polymorphism modulates circulating TNF-α levels as well as EDS independent of OSA severity in patients with CAD.

## 2. Materials and Methods

### 2.1. Study Participants

The methodology of the main RICCADSA trial has been published previously [27,28]. A total of 511 CAD patients with a history of percutaneous coronary intervention (PCI) or coronary artery bypass grafting (CABG) within 6 months prior to recruitment in the Skaraborg County of West Götaland, Sweden were included in the RICCADSA trial (Figure 1). The patient recruitment was carried out between 2005 and 2010. CAD patients with non-sleepy OSA (apnea–hypopnea index (AHI) ≥ 15/h) according to the home sleep apnea test (HSAT) at screening and Epworth Sleepiness Scale (ESS) < 10) were randomized to CPAP or no-CPAP groups. The sleepy OSA patients (ESS ≥ 10) were offered CPAP, and the CAD patients with an AHI < 5/h were included as no-OSA. For the current study, the blood samples were collected at the final visit during 2012/2013. In total, 326 out of 384 eligible participants were included in the current *TNF-α*-308G/A polymorphism study (Figure 1).

### 2.2. Study Oversight

The study protocol was approved by the Ethics Committee of the Medical Faculty of the University of Gothenburg (approval no. 207–05; 09.13.2005; amendment T744-10; 11.26.2010; amendment T512-11; 06.16.2011), and written informed consent was obtained from all patients. An additional ethics approval was obtained for the molecular analysis of the biomarkers (approval nr 814-17; 11.21.2017). The RICCADSA trial was registered with ClinicalTrials.gov (NCT 00519597) as well as with the national researchweb.org (FoU i Sverige—Research and development in Sweden; no. VGSKAS-4731; 29 April 2005).

### 2.3. Sleep Studies

The HSAT was conducted with the Embletta^®^ Portable Digital System device (Embla, Broomfield, CO, USA). As explained previously [27], the HSAT system included a nasal pressure detector and two respiratory inductance plethysmography belts for recording thoraco-abdominal movements and body position, in addition to a finger pulse-oximeter for recording heart rate and oxyhemoglobin saturation (SpO_2_). Apnea was defined as at least 90% cessation of airflow, and hypopnea as at least a 50% reduction in nasal pressure amplitude and/or in thoraco-abdominal movement for at least 10 s [29]. The total number of significant drops in SpO_2_ exceeding 4% from the immediately preceding baseline was also recorded, and the oxygen desaturation index (ODI) was determined as the number of significant desaturations per hour.

### 2.4. Epworth Sleepiness Scale

The ESS questionnaire [30] was assessed to determine the participants’ subjective daytime sleepiness. The ESS contains eight items regarding the risk of falling asleep under eight different situations in the past month. At least 10 out of 24 criteria were used to categorize patients as sleepy.

### 2.5. Comorbidities

As previously described [27], the anthropometrics, smoking habits and medical history of the study population were obtained from the medical records. Participants with a body-mass-index (BMI) ≥ 30 kg/m^2^ were categorized as obese, and abdominal obesity was defined as a waist-to-hip ratio (WHR) ≥ 0.9 for men and ≥0.8 for women [31].

### 2.6. Circulating TNF-α Levels

Blood samples were collected after an overnight fasting using ethylenediaminetetraacetic acid and serum tubes in the morning (07:00–08.00 am). Tubes were centrifuged and the plasma/serum samples were aliquoted and stored at −70 °C. TNF-α levels were analyzed in undiluted plasma samples using commercially available MILLIPLEX MAP (based on Luminex technology) human serum adipokine assay kits according to the manufacturer’s instructions (Merck Millipore, Burlington, MA, USA). The minimum detectable concentration (assay sensitivity) for TNF-α was 0.14 pg/mL. The TNF-α concentration in all samples (undiluted) was within the standard curve, ranging from 0 to 10,000 pg/mL. As previously described [32], the intra-assay and inter-assay variabilities (generated from the mean of the percentage coefficient of variability from multiple reportable results across two different concentrations of the samples in one experiment, or from two results each for two different concentrations of samples across several different experiments) were 1.4–7.9% and <21%, respectively.

### 2.7. TNF-α-308G/A (SNP Rs1800629) Genotyping

Venous blood samples were drawn into EDTA-containing tubes for DNA isolation. Genomic DNA was extracted from the whole blood using the PAXgene Blood DNA Kit (PreAnalytiX; Qiagen, Humbrechtikon, Switzerland). The quality and concentration of DNA samples were determined using a nano drop photometer (NanoDrop 2000; Thermo Scientific, Waltham, MA, USA) and DNA samples were stored at –80 °C until further use.

The genotyping of SNP Rs1800629 in the *TNF-α* gene promoter region (-308G/A) was performed by polymerase chain reaction–restriction fragment length polymorphism (PCR–RFLP). The DNA fragment harboring the *TNF-α*-308G/A SNP Rs1800629 was PCR-amplified using the forward primer 5′-AGGCAATAGGTTTTGAGGGCCAT-3′ and reverse 5′-TCCTCCCTGCTCCGATTCCG-3′ (product size 107 bp), as reported previously [13]. The total PCR volume was 25 μL using 50 ng DNA as a template according to the manufacturer’s protocol (PCR Master Mix, 2X, ThermoFisher, Waltham, MA, USA). PCR amplification was performed in a Biometra and BioRad thermocycler during 35 cycles of initial denaturation at 95 °C for 1 min, 35 cycles of denaturation at 95 °C for 30 s, annealing at 60 °C for 30 s, extension at 72 °C for 1 min, and final extension at 72 °C for 7 min. 

The amplicons were digested by 1 U of NcoI restriction enzyme (NcoI 10 U/μL, ThermoFisher) according to the manufacturer’s protocol. Restriction enzyme reactions were incubated at 37 °C for 4 h, and the enzyme was subsequently inactivated at 65 °C for 20 min. The expected fragment sizes after restriction enzyme digestion were 87 bp and 20 bp in the presence of the G allele (restriction site present) and 107 bp in the presence of the A allele (restriction site absent). The band pattern and intensity of all the produced bands were determined using the Fragment Analyzer 5200 (Agilent) by the DNF-905 dsDNA reagent kit (1–500 bp, Agilent), according to the manufacturer’s instructions (see Appendix A).

To ensure the accuracy of the amplified PCR fragment as well as the produced RFLP genotypes, PCR products of 16 randomly selected samples were subjected to Sanger sequencing (Eurofin Genomics) (Appendix A). The obtained sequences were subjected to a blast search, and the recorded genotypes were compared to those obtained from RFLP results for these 16 samples.

### 2.8. Statistical Analysis

For descriptive statistics, variables were reported as a median with interquartile ranges (IQRs) for continuous variables, and as a percentage for categorical variables. The Shapiro–Wilk test was used to test the normality assumption of the current data for all variables. The baseline differences between the three groups stratified by the *TNF-α*-308G/A gene polymorphism were tested by the Kruskal–Wallis test for continuous variables and by the Chi-square test for the categorical data. The Hardy–Weinberg equilibrium was tested by using the formula (p2 + 2pq + q2 = 1) for genotype distribution across all CAD subgroups. A univariate linear regression analysis was performed to test the association between circulating TNF-α levels and age, sex, ESS, BMI, WHR, AHI, ODI, OSA and comorbidities as well as *TNF-α* genotypes (coded as GG = 0, GA = 1, and AA = 2) and *TNF-α* alleles (coded as G = 0, and A = 1), respectively. Two distinct multivariate linear regression models were conducted to associate circulating TNF-α levels with the variable *TNF-α* genotypes and *TNF-α* alleles. Both multivariate models included the same significant covariates based on the univariate analysis as well as the variables age, BMI and sex in order to adhere to the recent guideline [33]. Moreover, a binary logistic regression analysis was performed to determine the variables associated with EDS. Age, sex, obesity, OSA severity categories, circulating TNF-α levels and *TNF-α* alleles were entered into the multivariate model. All statistical tests were two-sided, odds ratios (ORs) with 95% confidence interval (CI) were reported, and a *p*-value < 0.05 was considered significant. Statistical analyses were performed using SPSS^®^ 26.0 for Windows^®^ (SPSS Inc., Chicago, IL, USA).

## 3. Results

The study population consisted of 326 participants (mean age 64.4 ± 8.6 years; male, 84%). The comparison of the amplified PCR fragments with RFLP genotyping and with Sanger sequencing in 16 random samples showed consistent results.

As shown in Table 1, the patients from the entire RICCADSA cohort, whose blood samples were not eligible at the final follow-up visit during 2012–2013 (*n* = 186), were slightly younger (*p* = 0.039), had a higher proportion of current smokers (*p* = 0.042) and lung disease (*p* = 0.041) and were slightly less frequent users of statins (*p* = 0.028) at baseline compared to the patients who participated in the current study. Gender, BMI, obesity, OSA, AHI, ODI, ESS score and EDS as well as circulating TNF-α levels were similar (Table 1).

As presented in Table 2, baseline demographic and clinical characteristics were similar across the TNF-α genotypes. The proportion of obese individuals tended to be greater in the *TNF-α*-AA genotype, whereas EDS was more common for the *TNF-α*-GG genotype.

As illustrated in Figure 2A, *TNF-α*-GG was the most prevalent genotype, and *TNF-α*-AA was the least frequent, both in the OSA and no-OSA groups and in the sleepy and non-sleepy OSA subgroups. The allele distribution was similar across all groups, with *TNF-α*-308G the most common allele in the entire study cohort (Figure 2B). The *TNF-α*-308A allele was found at a rate of 23% in OSA vs. 27% in no-OSA (n.s). In total, the genotype distribution of the study population across the subgroups did not deviate from the Hardy–Weinberg equilibrium.

In the entire study population, the median values of the circulating TNF-α levels were higher among CAD patients with OSA (5.05 pg/mL (IQR 3.43–6.94)) compared to 4.62 pg/mL (IQR 3.06–6.31) in CAD patients with no-OSA, but the difference was not statistically significant. The corresponding values were 5.05 pg/mL (IQR 3.23–6.93) in the sleepy OSA phenotype compared to 5.15 pg/mL (IQR 3.67–6.99) in the non-sleepy OSA phenotype (n.s).

Among the variables tested in univariate linear regression analyses, ODI in addition to *TNF-α* genotypes from GG to GA and GA to AA as well as the *TNF-α*-308A allele were significantly associated with the circulating TNF-α levels (Table 3).

In the multivariate models, TNF-α genotypes were significantly related with the TNF-α levels, whereas the significance of ODI remained in the model with WHR but not with BMI (Table 4). Both the TNF-α-308A allele and ODI were significantly associated with the circulating TNF-α levels independent of age, sex, BMI and WHR (Table 4).

Further univariate logistic regression analyses revealed age, obesity, OSA severity and the *TNF-α*-308A allele to be the meaningful associates of the EDS phenotype. The CAD patients carrying the *TNF-α*-308A allele remained significantly related with the reduced risk for EDS (OR 0.64, 95% CI 0.41–0.99; *p* = 0.043) independent of age, sex, obesity, OSA severity and the circulating TNF-α levels in the multivariate model (Figure 3).

## 4. Discussion

This study indicated that *TNF-α*-308G/A gene polymorphism was independently associated with circulating TNF-α levels, with a significant increase across the genotypes from GG to GA and GA to AA. Moreover, the *TNF-α*-308A allele was associated with a significant risk reduction for the occurrence of EDS in adults with CAD, independent of the confounding factors of age, sex, obesity, OSA severity and circulating TNF-α levels.

To the best of our knowledge, this is the first study addressing the association of *TNF-α*-308G/A polymorphism with circulating TNF-α levels as well as with EDS in an adult cardiac population with concomitant OSA. An association between the *TNF-α*-308A allele and OSA susceptibility has been suggested previously. Riha et al. reported a significant difference in the -308A allele frequency in a British population study (*n* = 206), where the -308A allele occurred in 28% of OSA subjects compared to 18% in the healthy controls [24]. Similar results were reported in an obese Asian Indian population (*n* = 207), where the -308A allele frequency in the OSA cohort was 28%, and the frequency was 13% in the control group [22]. Neutral results were also reported; the -308 A allele distribution was 14% in OSA vs. 12% in no-OSA in a Polish cohort (*n* = 179) [21], and 13% in OSA vs. 11% in no-OSA in a Turkish population (*n* = 111) [34]. However, according to meta-analyses that included studies with differences in cohort characteristics, the *TNF-α*-308G/A polymorphism has been shown to be significantly associated with OSA [35,36].

In the current cohort, the *TNF-α*-308A allele was found among 23% of the patients with OSA vs. 27% in the no-OSA group. One explanation for the higher -308A allele frequency in the no-OSA group in our study might be the confounding effect of other comorbidities such as obesity, hypertension and diabetes mellitus and/or CAD per se, given that those individuals without OSA were not healthy controls. Similar to OSA, CAD is an inflammatory condition mediated by the activity of pro-inflammatory cytokines including TNF-α [37].

The role of *TNF-α* gene polymorphism in CAD pathogenesis has been extensively investigated with inconsistent results. Some researchers have reported that *TNF-α*-308G/A polymorphism is implicated in CAD development [37,38], whereas some others reported no evidence in this context [39]. It is suggested that the implication of the polymorphism in CAD pathogenesis differs between geographical ethnicities, and there was no significant association of *TNF-α*-308G/A polymorphism with the development of CAD according to a recent meta-analysis [26].

Circulating TNF-α levels have been postulated to be elevated in individuals with OSA compared to controls [25,40]. In a meta-analysis, it has also been suggested that circulatory TNF-α levels are related with the severity of OSA [40]. We also found higher levels of circulating TNF-α levels among patients with OSA, but the difference was not statistically significant when categorizing the participants based on the AHI values. In the regression analysis, not AHI but ODI was significantly related with circulating TNF-α levels. As previously stated in a baseline cross-sectional analysis of the entire RICCADSA population [32], intermittent hypoxemia, rather than the number of apneas and hypopneas, seemed to be primarily related with increased inflammatory activity among patients with CAD and concomitant OSA.

As mentioned above, circulating levels of inflammatory markers predict future cardiovascular events in general and cardiac populations [8,9]. It has also been suggested that inflammation can be a predisposing factor for OSA [13,14,15], so this association could be bidirectional. The normalization of circulatory levels of inflammatory markers following the CPAP treatment of OSA patients in observational studies was supportive of a causative link between OSA and systemic inflammation [12]. However, in our previous randomized controlled trial of the effect of CPAP on the inflammatory markers, including hs-CRP, IL-6, IL-8 and TNF-α in the non-sleepy arm of the RICCADSA cohort, only IL-6 levels decreased after one year, both in the CPAP and no-CPAP arms [41]. This was probably suggestive of a natural course of improvement in the cardiac disease rather than any effect of CPAP treatment. Additional studies addressing the associations of gene polymorphisms, including TNF-α and IL-6, with the changes in circulatory levels of those cytokines from baseline in response to CPAP treatment in the current cohort are in progress, which may hopefully give more insights into the complex relationship between systemic inflammation and OSA in patients with CAD.

The *TNF-α*-308A allele is suggested to promote a two-fold increase in TNF-α transcription activity [42]. We found a significant increase in the circulating TNF-α levels across the *TNF-α*-308 genotypes from GG to GA and GA to AA. TNF-α is a mediator in the sleep regulatory system, and the fragmented sleep pattern associated with OSA is believed to increase the circulatory levels of TNF-α [17,40]. Thus, sleepy OSA subjects would be expected to experience a more fragmented sleep pattern and would subsequently be expected to have higher circulatory TNF-α levels, which was not the case in our study population. This controversy might be partly explained by the modulatory role of the *TNF-α*-308A allele on the circulating TNF-α levels as well as its protective role against EDS in patients with CAD and concomitant OSA. Moreover, interactions of several SNPs in the promoter region of TNF-α may collectively affect the expression level, and thus the subjects in the present cohort could be carriers of SNPs that also affect the gene activity, which were not covered by the current study. It is worthy of note that there are also other recent genetic studies that have found polymorphisms in the IL-18 receptor gene region, which was associated with SpO_2_ at the genome-wide significance level [43].

Given that only 326 of 511 participants in the original clinical trial were included in the current study, one may argue that the exclusion of participants might have led to biased estimates. However, the baseline characteristics of the patients that were not included in the current study did not differ from the included patients except for slight variations in age, current smoking, lung disease and statin use at baseline, which probably do not have any clinical importance in the context of the outcomes of the current study. In the literature, there have been reports suggesting that statins may influence sleep quality [44], and the slight difference in the proportion of patients regarding statin treatment in the current study population probably had no influence on the outcomes either. The most important factors of gender, BMI, obesity and occurrence of OSA as well as OSA severity in terms of AHI and ODI, distribution of sleepy vs. non-sleepy OSA phenotypes, ESS score and circulating TNF-α levels were similar in both groups, which confidently supports the concept that the current study population was representative of the whole RICCADSA cohort.

The current study has certain limitations. Firstly, the power estimate for the entire RICCADSA cohort was conducted for the primary outcome and not for the secondary outcomes assessed in this study for the comparison of genotypes and alleles across the subgroups. However, the total number of the participants was comparable with the number of participants reported in the aforementioned studies. Secondly, EDS was categorized based on the ESS threshold, which may not be precise in a CAD population, and the clinical reproducibility of ESS has been shown to be variable when administered sequentially [45]. Nevertheless, this questionnaire is a largely accepted tool used in clinical cohorts [30], and objective measurements such as the Multiple Sleep Latency Test [46] are time-consuming and not realistic to conduct in large-scale cardiac populations. Thirdly, our results are not generalizable to adults with OSA in the general population or in sleep clinic cohorts as we did not have a control group without CAD. This notwithstanding, our results in this well-defined revascularized CAD cohort may help to identify patients who are at risk of cardiovascular adverse outcomes and persistent EDS in spite of an optimal treatment of OSA with CPAP or other treatment modalities. Further longitudinal analysis of the current cohort would hopefully give some additional insights into the molecular basis of these differences in cardiac patients with concomitant OSA.

## 5. Conclusions

In brief, *TNF-α*-308G/A gene polymorphism and OSA severity in terms of ODI were independently associated with circulating TNF-α levels in the current CAD cohort. The *TNF-α*-308A allele appears to modulate circulatory TNF-α levels and mitigate EDS independent of age, sex, obesity and OSA severity in adults with CAD.

## Figures and Tables

**Figure 1 jcm-10-03413-f001:**
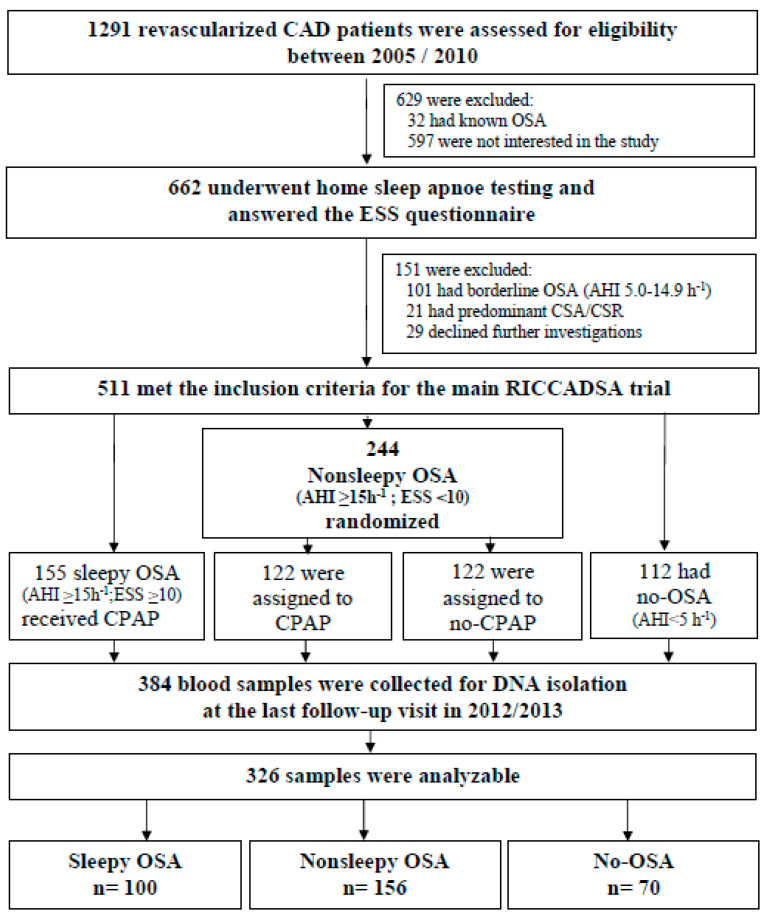
Flow of patients through the study. Abbreviations: AHI, apnea–hypopnea index; CAD; coronary artery disease; CPAP, continuous positive airway pressure; CSA-CSR, central sleep apnea–Cheyne Stokes respiration; ESS, Epworth Sleepiness Scale; OSA, obstructive sleep apnea; RICCADSA, Randomized Intervention with CPAP in Coronary Artery Disease and Sleep Apnea.

**Figure 2 jcm-10-03413-f002:**
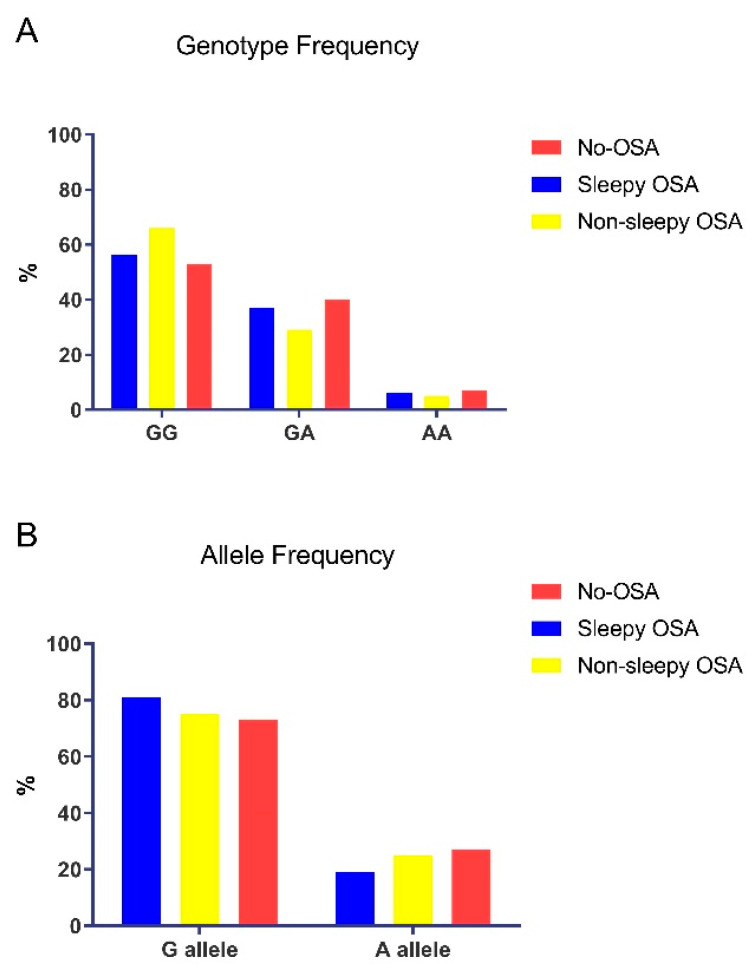
(**A**) Genotype frequency of the *TNF-α*-308G/A promoter polymorphism and (**B**) allele frequency of the *TNF-α*-308G/A promoter polymorphism in the study cohort.

**Figure 3 jcm-10-03413-f003:**
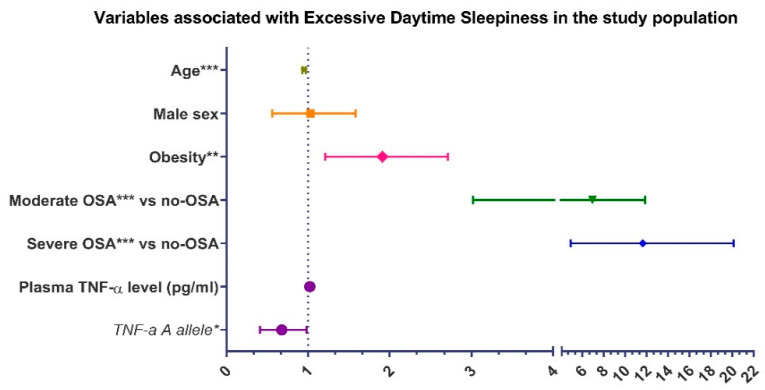
Variables associated with excessive daytime sleepiness in the study population in the multivariate logistic regression analysis. Abbreviations: TNF-α = tumor necrosis factor alpha; OSA = obstructive sleep apnea. * *p* < 0.05, ** *p* < 0.01, *** *p* < 0.001.

**Table 1 jcm-10-03413-t001:** Baseline demographic and clinical characteristics of the entire RICCADSA cohort with vs. without eligible blood samples for the genetic analysis.

	Eligible *n* = 326	Not Eligible *n* = 186
Age, years	64.4 (59.7–70.9)	62.1 (57.3–69.7)
Male sex, %	84.0	82.7
BMI, kg/m^2^	27.8 (25.6–30.0)	28.0 (25.4–31.0)
Obesity, %	24.5	32.4
WHR	0.95 (0.92–1.00)	0.97 (0.92–1.00)
Abdominal obesity, %	90.4	90.5
Current smoking, %	16.0	23.2
ESS score	7.0 (4.0–10.0)	7.0 (4.0–10.0)
EDS (ESS score ≥ 10), %	32.5	30.8
OSA (AHI ≥ 15 events/h), %	78.5	77.3
AHI, events/h	22.1 (15.6–34.3)	20.7 (15.4–30.8)
ODI, events/h	11.1 (4.4–21.3)	10.0 (3.4–19.5)
Hypertension, %	56.7	60.5
Lung disease, %	7.1	12.4
Diabetes, %	20.2	25.4
Stroke, %	6.2	7.6
Depression, %	3.5	7.3
Statin use, %	95.9	91.2
Plasma TNF-α (pg/mL)	5.01 (3.31–6.85)	4.82 (3.33–7.21)

Continuous data are presented as median and 25–75% quartiles. Categorical data are presented as percentages. Abbreviations: AHI = apnea–hypopnea index; BMI = body mass index; EDS = excessive daytime sleepiness (ESS score ≥ 10); ESS = Epworth Sleepiness Scale; TNF-α = tumor necrosis factor alpha; ODI = oxygen desaturation index; OSA = obstructive sleep apnea; WHR = waist hip ratio.

**Table 2 jcm-10-03413-t002:** Baseline demographic and clinical characteristics of the study population.

	GG *n* = 191	GA *n* = 115	AA *n* = 20
Age, years	64.4 (59.0–70.9)	64.8 (60.3–71.1)	66.6 (59.2–71.8)
Male sex, %	84.8	82.6	85.0
BMI, kg/m^2^	27.5 (25.6–30.1)	27.5 (25.7–29.7)	29.0 (26.3–30.9)
Obesity, %	25.1	22.6	30.0
WHR	0.95 (0.91–0.99)	0.95 (0.92–1.00)	0.97 (0.92–1.00)
Abdominal obesity, %	89.5	90.3	100
Current smoking, %	16.2	15.7	15.0
ESS score	8.0 (4.0–10.7)	7.0 (4.0–10.0)	7.0 (4.0–10.0)
EDS (ESS score ≥ 10), %	36.6	27.0	25.0
OSA (AHI ≥ 15 events/h), %	80.6	75.7	75.0
AHI, events/h	22.5 (15.9–34.9)	22.2 (9.9–33.1)	21.7 (4.9–39.2)
ODI, events/h	11.2 (4.8–21.7)	10.2 (3.9–20.2)	12.5 (3.9–26.4)
Hypertension	55.5	57.4	65.0
Lung disease, %	8.9	5.2	0
Diabetes, %	21.5	17.4	25.0
Stroke, %	6.8	5.3	5.0
Depression, %	4.3	1.8	5.0
Statin use, %	97.3	94.6	89.5
Plasma TNF-α (pg/mL)	4.95 (3.25–6.85)	5.26 (3.90–7.60)	5.05 (2.87–6.36)

Continuous data are presented as median and 25–75% quartiles. Categorical data are presented as percentage. Abbreviations: AHI = apnea–hypopnea index; BMI = body mass index; EDS = excessive daytime sleepiness (ESS score ≥10); ESS = Epworth Sleepiness Scale; TNF-α = tumor necrosis factor alpha; ODI = oxygen desaturation index; OSA = obstructive sleep apnea; WHR = waist hip ratio.

**Table 3 jcm-10-03413-t003:** Univariate analyses of the variables associated with circulating TNF-α levels.

	Standardized β	95% Confidence Interval for		*p* Values
		Lower Bound	Upper Bound	
*TNF-α* A Allele	**0.09**	0.07	1.90	**0.034**
TNF-α Genotypes *	0.13	0.13	1.90	**0.024**
Age, years	0.07	−0.02	0.11	0.201
Male sex	0.06	−0.67	2.31	0.279
BMI, kg/m^2^	0.03	−0.10	0.18	0.607
Obesity	−0.00	−1.27	1.25	0.988
WHR	−0.03	−2.08	1.23	0.613
AHI, events/h	0.08	−0.01	0.05	0.180
ODI, events/h	0.11	0.00	0.08	**0.049**
OSA, (AHI ≥ 15 events/h)	0.02	−1.10	1.55	0.739
ESS score	0.07	−0.5	0.23	0.188
EDS, (ESS score ≥ 10)	0.03	−0.82	1.49	0.570
Current smoking	0.06	−0.70	2.27	0.300
Hypertension	0.07	−0.39	1.80	0.207
Diabetes	−0.00	−1.38	1.33	0.975
Stroke	−0.02	−2.51	1.78	0.729
Lung disease	0.00	−2.11	2.21	0.963
Depression	−0.04	−4.14	1.90	0.466

Abbreviations: AHI = apnea–hypopnea index; BMI = body mass index; EDS = excessive daytime sleepiness (ESS score ≥10); ESS = Epworth Sleepiness Scale; TNF-α = tumor necrosis factor alpha; ODI = oxygen desaturation index; OSA = obstructive sleep apnea; WHR = waist hip ratio. * GG = 0, GA = 1, AA = 2. The bold typeface indicates significantly different (*p* < 0.05).

**Table 4 jcm-10-03413-t004:** Multivariate analysis of the variables associated with circulating TNF-α levels.

		Standardized β	95% Confidence Interval for		*p* Values
			Lower Bound	Upper Bound	
Model 1	Genotypes *	0.12	0.07	1.99	0.035
	ODI	0.02	0.00	0.09	0.028
	Age	0.03	−0.04	0.10	0.962
	Male sex	0.80	−0.80	2.40	0.981
	WHR	−0.52	−2.20	1.16	0.541
Model 2	Genotypes *	0.12	0.08	1.94	0.034
	ODI	0.11	−0.00	0.09	0.072
	Age	0.06	−0.03	0.11	0.294
	Male sex	0.04	−0.99	2.10	0.484
	BMI	−0.00	−0.17	0.15	0.931
Model 3	*TNF-α* A allele	0.08	0.00	2.00	0.048
	ODI	0.14	0.02	0.09	0.001
	Age	0.06	−0.02	0.09	0.170
	Male sex	0.06	−0.34	2.00	0.162
	WHR	−0.33	−1.68	0.72	0.432
Model 4	*TNF-α* A allele	0.08	0.02	1.97	0.045
	ODI	0.12	0.12	0.08	0.008
	Age	0.07	−0.01	0.09	0.124
	Male sex	0.04	−0.55	1.72	0.312
	BMI	−0.00	−0.12	0.12	0.989

Abbreviations: AHI = apnea–hypopnea index; ESS = Epworth Sleepiness Scale; ODI = oxygen desaturation index; TNF-α = tumor necrosis factor alpha; * GG = 0, GA = 1, AA = 2. The bold typeface indicates significantly different (*p* < 0.05).

## Data Availability

Individual participant data that underlie the results reported in this article can be obtained by contacting the corresponding author, yuksel.peker@lungall.gu.se.

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
