# Peer review of "Association of TNF-α (-308G/A) Gene Polymorphism with Circulating TNF-α Levels and Excessive Daytime Sleepiness in Adults with Coronary Artery Disease and Concomitant Obstructive Sleep Apnea"

_jcm, 2021, doi:10.3390/jcm10153413_

Round 1
Reviewer 1 Report
I have read the paper of Behboudi et al. with great interest. The authors took the interesting topic of the association between TNF-α gene polymorphism with excessive daytime sleepiness in adults with coronary artery disease and concomitant OSA. In my opinion, there are some elements, which should be changed:
Major:
- The introduction should be rewritten in a more consistent manner. Moreover, some aspects of the relationship between OSA and CAD should be added (fe. doi:10.3389/fneur.2018.00635)
- The introduction should be ended with the hypothesis of the study.
- Materials and methods: can the administered drugs affect sleep quality? It should be stated in the context of, e.g.: 10.1016/j.smrv.2020.101380
Minor:
- Materials and methods: You stated that: "Quality and concentration of DNA samples were determined using the nano drop photometer (NanoDrop 2000; Thermo Scientific) and DNA samples 151 were stored in – 80 degrees until further use.". Tell us about the minimal quality parameters for further processing. Why did you store DNA samples at -80 degrees, when the temperature of approximately -20 is enough for DNA storage?
- The current quality of figures 2 and 3 seems to be bad. Please provide them with a better resolution.
- The following paper: 10.1111/nmo.13978 may be considered as worth be discussed.
Reviewer 2 Report
- Although the results were not consistent with the hypothesis and some prior evidence, this study was overall well conducted and the manuscript was clearly written. One possible explanation that the authors did not discuss is the potential for selection bias, given that only 326 of 511 participants in the original clinical trial were included in the current study. Do the authors think the exclusion of participants may be nonrandom that have led to biased estimates?
- The authors should also recognize the limitations of using ESS to assess daytime sleepiness (e.g., PMID: 29734985, 17557491), which may to some extent contribute to the observed results.
- A growing number of studies suggest that inflammation can be a predisposing factor for OSA (PMID: 33529772, 15151922, 12186824), not just the consequence of these disorders. Given this is a cross-sectional study, the authors should incorporate these studies and acknowledge this possibility early in the introduction and further discuss the potential bidirectional associations between inflammation and OSA.
- There are more recent GWAS studies linking some inflammation-related SNPs with OSA traits, such as overnight hypoxia. For example, the study by Cade et al. (PMID: 30990817) found genetic polymorphisms in the IL-18 receptor gene region was associated with SpO2 at the genome-wide significance level.
- Figure 2 and 3 can be moved to supplemental materials, as they are not directly related to the research question.
Reviewer 3 Report
The authors report here as an ancillatory study of the RICCADSA trial that TNF-α -308G/A gene polymorphism is independently associated with the circulating TNF-α levels with a significant increase across the genotypes
from GG to GA, and GA to AA in a CAD population.
Moreover, the TNF-? -308A allele was associated with a significant risk reduction for having EDS in adults with CAD, independent of the confounding factors age, sex, obesity, OSA severity and the circulating TNF-α levels.
These results are important because they will allow a better understanding of the broad response phenotype in terms of excessive daytime sleepiness in our apneic coronary patients and this independently of the severity of the OSA (very often). There is no doubt about the quality of the study and the methodology. We ask the authors for more details on the serums not available for analysis and if this changes the representativeness of the population explored.
The authors focus their research on TNF alpha polymorphism and expression but little is known about the cytokine inflammatory status of this same population. Was IL6 in particular measured? The mainstream of the inflammatory cascade associated with chronic intermittent hypoxia needs to be addressed a little better in the discussion chapter
This is a very nice work.
